# High-resolution computed tomography with scattered X-ray radiation and a single pixel detector
A. Ben-Yehuda [1], O. Sefi[1], Y. Klein[1], H. Schwartz[1], E. Cohen [2], R. H. Shukrun [1,3] & S. Shwartz [1] ✉

X-ray imaging is a prevalent technique for non-invasively visualizing the interior of the human body and other opaque samples. In most commercial X-ray modalities, an image is formed by measuring the X-rays that pass through the object of interest. However, despite the potential of scattered radiation to provide additional information about the object, it is often disregarded due to its inherent tendency to cause blurring. Consequently, conventional imaging modalities do not measure or utilize these valuable data. In contrast, we propose and experimentally demonstrate a high resolution technique for X-ray computed tomography (CT) that measures scattered radiation by exploiting computational ghost imaging (CGI). We show that the resolution of our method can exceed 500 μm, which is approximately an order of magnitude higher than the typical resolution of X-ray imaging modalities based on scattered radiation. Our research reveals a promising technique for incorporating scattered radiation data in CT scans to improve image contrast and resolution while minimizing radiation exposure for patients. The findings of our study suggest that our technique could represent a significant advancement in the fields of medical and industrial imaging, with the potential to enhance the accuracy and safety of diagnostic imaging procedures.

Commercial X-ray scanners utilize the same physical principle that Wilhelm Roentgen demonstrated in 1895 when he used X-rays to image the hand of his wife[1]. The X-rays passing through the object are absorbed to varying degrees by the different structures, creating an image that shows the internal composition of the object. In medical imaging, for instance, denser structures such as bones absorb more X-rays and appear white on the image, while softer tissues, like muscles and inner organs, absorb fewer X-rays and appear darker. This very simple concept has proven useful and robust, making X-rays one of the most valuable medical and industrial imaging modalities. However, it became apparent soon after X-rays were first used for imaging that the image quality was significantly degraded in many practical scenarios where the density variations of the organs were small or when the volume of the object was significant[2–5]. When X-rays interact with the electrons in the object, they are not only absorbed but are also scattered, introducing significant blurring and distortions of the images. To partially mitigate the impact of scattered radiation on the image quality, grids or collimators are used, but these tools also increase the radiation dose leading to a significant increase in the risk of radiation damage[6–11]. Notwithstanding

the successful application of those devices, their ability to reduce scattering is limited and might be insufficient for scenarios where the contrast of the image is low or when a detailed image of the object is required. Despite numerous endeavors to improve image quality and reduce radiation dose, scattering remains a persistent challenge[12–22], especially in CT scans since they target large volumes and low contrasts.

Compton scattering in human tissues is stronger than the absorption for the photon energy range typically used for CT scans (80–150 keV), as illustrated in Fig. 1[23], which shows the absorption (the photoelectric effect) and the Compton scattering (the incoherent scattering) in cortical bones as a function of the photon energy of the X-ray beam. This plot clearly shows that current X-ray imaging techniques leave substantial amount of energy unused, necessitating higher doses to achieve adequate image quality[24,25].

Here we propose and experimentally demonstrate a technique for CT that measures the scattered X-ray radiation by utilizing a ghost imaging approach. We combine computational ghost imaging (CGI)[26–28] with a recently developed deep learning algorithm[29], and an advanced CT image reconstruction toolbox[30] to reconstruct a high resolution three-dimensional

[1]Physics Department and Institute of Nanotechnology and Advanced Materials, Bar-Ilan University, Ramat Gan 52900, Israel. [2]Faculty of Engineering and Institute of Nanotechnology and Advanced Materials, Bar-Ilan University, Ramat Gan 52900, Israel. [3]Radiation Safety Department, Soreq Nuclear Research Center, Yavne 81800, Israel. ✉e-mail: Sharon.shwartz@biu.ac.il

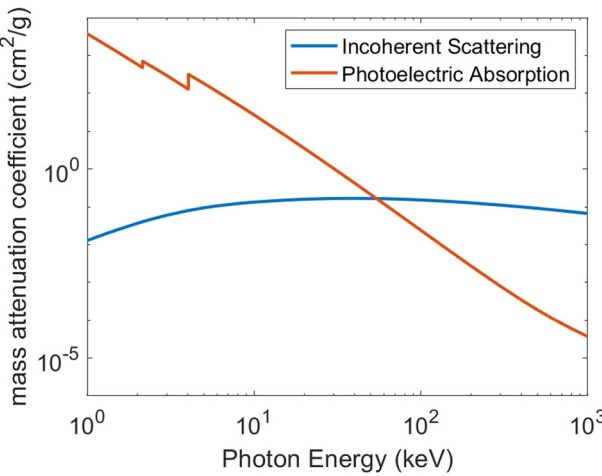

**Fig. 1 | Photon energy dependence of the cross-sections of photo absorption (orange) and Compton scattering (blue) in cortical bones.** The incoherent scattering is stronger than the photoelectric absorption for photon energy range of 80–150 keV.

**Table 1 | The parameters used in the GIDC code (also provided as Python code Supplementary Fig. 8)**

| | |
|---|---|
| Image width | 128px |
| Image Hight | 128px |
| Sampling Rate | 0.211 |
| Batch size | 1 |
| Learning rate | 0.002 |
| TV strength | $1 * 10^{-3.9}$ |
| Number of patterns | 3468 |
| Number of optimization steps | 1501 |

(3D) CT image of a part of a large mammal bone. Our results unequivocally demonstrate the capability of our method to provide a resolution of at least 500 μm with scattered X-ray radiation. This performance surpasses that of commercially available X-ray imaging modalities based on scattered raiation[31,32], and significantly enhances image contrast compared to traditional transmission-based X-ray imaging modalities.

We chose to employ CGI, which is a form of single-pixel imaging method since it has been shown to be more robust to scattering than standard imaging methods[33,34]. This is because in CGI the spatial information is derived from the spatially modulated beam, which interacts with the object, rather than from the detector pixels. While X-ray CGI has been previously demonstrated with single-pixel detectors for collecting transmitted[35–39], fluorescent[40], and refracted radiation[41], we apply this technique to incoherent (Compton) scattered X-ray radiation. This approach is akin to the method proposed by A. M Kingston et al. for imaging of scattered neutrons[42].

It is worth mentioning that although cameras based on the Compton effect are available[31,32], their resolution is poor due to the tendency of scattered radiation to blur. As a result, they are not suitable for medical imaging, unlike our proposed method which provides higher resolution and better image quality.

In CGI, the input beam is spatially modulated to produce intensity patterns on the object. The signal from the object, which can be the intensity transmitted, refracted, or scattered, is then detected by a single-pixel detector and registered by our acquisition system. The measurement is repeated for different patterns and the image is reconstructed by solving an inverse problem defined by the equation:

$$Ax = S, \tag{1}$$

where the mask patterns are represented by the matrix $A$, for which every row is a single illumination structure, the vector $x$ is the response function of the object (either the transmission, the reflection, or scattering), and $S$ is the detected signal. The reconstruction of the scatter image is done by solving the equation for the vector $x$.

To expedite the reconstruction process and minimize measurement time, we employed a reconstruction algorithm based on the technique of Ghost Imaging using Deep neural network Constraint (GIDC). The algorithm, for the technique was developed to leverage an untrained, self-supervised deep neural network (DNN) to generate far-field super-resolution with visible light[29]. In the present work we were able to modify GIDC (Table 1) to work with X-ray scatter radiation. The approach is based on the

concept that a DNN with randomly initialized weights can recover an image more accurately by adding a conventional regularization term, such as the total-variation (TV)[43]. The minimization of the loss function of GIDC is crucial in order to obtain a high-quality image. The weights of the DNN are adjusted in each iteration with the constraint of the pre-determined regularization parameter. The loss function is reformulated as follows:

$$T_{\varphi*} = argmin_{\varphi}\left|AT_{\varphi}(x) - S\right|^2 + \tau\mathfrak{T}\left[T_{\varphi}(x)\right], \tag{2}$$

where $T_{\varphi*}$ is the DNN, defined by a set of weights and biases parameters $\varphi$. The objective of GIDC is to find an optimal configuration $\varphi^*$ for the neural network, which effectively constrains the network output to produce a 1D sequence $\widetilde{S} = AT_{\varphi}(x)$, in accordance with the GI image formation physics as described by Eq. (1). This reconstructed sequence should closely resemble the experimentally acquired bucket signal $S$, where $x$ represents the response function of the object. The symbol $\mathfrak{T}$ represents the TV operator that works on the reconstructed image while the symbol $\tau$ denotes the regularization parameter. This parameter allows us to determine the degree of sparsity to enforce on the minimized term. The following procedure provides a two-dimensional image that contains 16,384 pixels constructed using 3468 samplings. We chose this number of samplings because we observed negligible improvement of the image quality when the number of samplings exceeded this value (see Supplementary Note 5 and Supplementary Fig. 6).

To perform a 3D image reconstruction using tomography techniques, it is essential to obtain projections of the object from various angles. We successfully accomplish this for both transmission and scatter images (see Supplementary note 1). In the case of transmission, we captured the images directly with a flat panel detector. For the scatter images we employed our CGI technique. We repeated the procedure multiple times, rotating the object each time, which allowed us to reconstruct 3D images from 28 different angles. This number of angles proved sufficient for our relatively simple object. For more complex objects, it is likely that more angles will be required, but this can be achieved as in any standard CT scan.

We implemented the GI with our scattered radiation scheme using the experimental setup illustrated in Fig. 2. It includes an X-ray source with parameters tuned to 80 kVp and 2 mA. To minimize scattering from the surrounding environment we employed 500 mm long circular collimator with a radius of 7.5 mm (not shown), positioned between the source and the object. The beam divergence after passing through the collimator was estimated to be 0.85°. We used a slit (not shown) to reduce the beam size at the object to $17 \cdot 19.5$ mm², which is comparable to the size of the object. The spatial modulation of the beam immediately before reaching the object was achieved by a mask comprised of absorbing silver features. These features have transverse dimensions of roughly 100 μm and thickness of about 1500 μm. The distance between the source and the mask is 1300 mm and the object is located 50 mm downstream of the mask. The object underwent 360° rotation during the measurements, facilitated by a rotation stage (not shown).

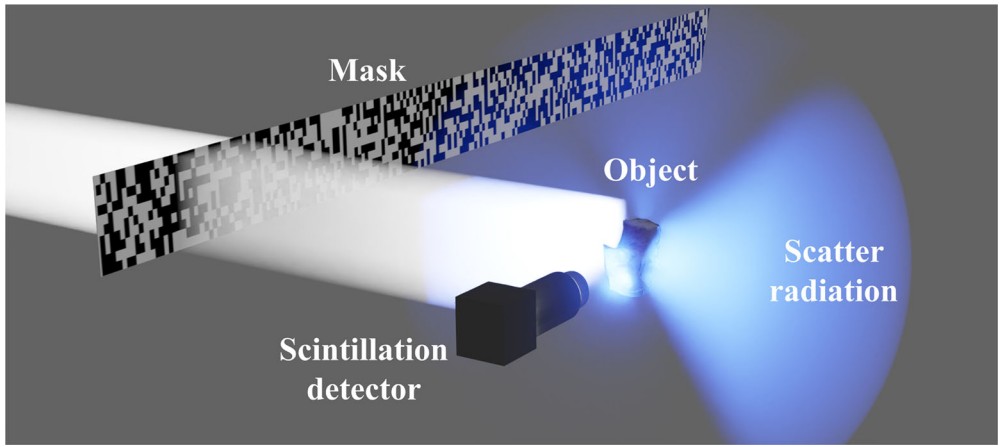

**Fig. 2 | Schematic of the experimental setup.** The silver mask modulates the beam, which irradiates the object (bone) with known random binary speckle patterns. The scattered X-ray radiation is then collected by a single pixel scintillation detector and the resulting signals are used to reconstruct a computational image from different angles as the object is rotated using a rotation stage (not shown).

The mask patterns are known, randomly distributed binary patterns with transmission-absorption ratio of 1:1, that are irradiated onto the object. High resolution X-ray GI techniques require changing the incident beam for having high contrast and small feature size which can be challenging at high photon energies, as photon penetration depth increases. To overcome this challenge, we adopted a technology made for electrical circuits 3D printing as it can provide high aspect ratio, 100 µm printing resolution with silver (see Methods section). The scintillation detector is located 50 mm from the object. To demonstrate the reliability of our imaging method, the detector was mounted at ~90° relative to the input beam since the differential cross section given by the Klein-Nishina formula at this angle is small[44] (low scattering angle):

$$\frac{d\sigma}{d\Omega} = \frac{1}{2}r_e^2\left(\frac{\lambda}{\lambda'}\right)^2\left[\frac{\lambda}{\lambda'} + \frac{\lambda'}{\lambda} - \sin^2(\theta)\right], \qquad (3)$$

where $\frac{d\sigma}{d\Omega}$ is the differential cross section, $\lambda$ is the wavelength of incident X-ray photon, $\lambda'$ is the wavelength of scattered X-ray photon, $\theta$ is the scattering angle of the scattered photon and $r_e^2$ is the classical electron radius. We first calibrated the CT reconstruction algorithm[45] by adjusting to the variables which provide the best-quality 3D CT image, with 28 projections by utilizing transmission X-ray we acquired with a flat panel detector (Table 2). Next, we used these variables to reconstruct the 3D CT image with the corresponding projections we acquired from the scattered X-ray radiation.

To validate our method and to benchmark its efficiency, resolution, and sensitivity against the standard direct transmission CT reconstruction, we reconstructed 3D images of a bone using signals obtained from transmission negative images, scattering reconstruction images, and their normalized average.

## Results and discussion

In Fig. 3a–l we present the isosurface of the bone from the reconstructed images using all three methods and compare it to a LiDAR 3D image in Fig. 3m–p. Due to its resilience to image degradation caused by scattering, the scatter reconstruction reveals finer details on the surface of the object when compared to the transmission reconstruction, demonstrating the superior ability of our method to capture intricate object features. The theoretical resolution of our method is estimated by the autocorrelation width of the mask used to generate the patterns and the reconstruction algorithm[46,47] (see Supplementary Note 2 and Supplementary Fig. 2). In our case, this length is 155 µm. The spatial resolution of the images obtained by direct imaging with a flat panel detector is determined by the resolution of the detector, which we estimate to be about 500 µm, reflecting the blurring by the scintillation screen. The combined reconstruction reveals features which are blurred in the transmission reconstruction due to scattering. This suggests a new approach to eliminate the need for collimators after the object while maintaining high-quality images despite scattering. Adopting this approach has the potential to significantly reduce radiation exposure as collimators absorb a substantial amount of radiation, which represents lost information that could have been collected as demonstrated in our experiment.

It is important to note that unlike previous works on 3D reconstructions using single-pixel detectors with visible light[48,49], which recovered only the surface gradients to derive the 3D surface of the object, we, like A. M. Kingston et al. which achieved GI tomographic reconstruction of transmitted radiation with synchrotron radiation[50], reconstructed a 3D volume that contains information about both the internal parts and the surface of the object with a tabletop X-ray setup. This enables us to present tomogram slices from top to bottom of the object as is shown in Fig. 4a–i. We specifically focused on a small hole in the bone, whose size is ~2000 µm on one end and 1200 µm on the other and presented its cross sections in Fig. 4j, k. The images and cross sections reveal that the edge of the hole is nearly imperceptible in the transmission tomogram due to scattering. In contrast, the scatter tomogram remains resilient to this effect and clearly depicts the hole. This indicates that our method provides higher resolving power compared to standard transmission-based CT. To further explore the ability of our method to resolve fine details, we evaluated the reconstruction resolution by measuring the cutoff frequency of the Fourier ring

## Table 2 | Computational tomography (CT) reconstruction tool code parameters (also provided as MATLAB code in Supplementary Fig. 9)

| | |
|---|---|
| Projection angles | 0, 180, 90, 270, 45, 225, 315, 135, 22.5, 202.5, 67.5, 247.5, 112.5, 292.5, 157.5, 337.5, 355, 175, 348, 168, 353, 173, 5, 185, 12, 192, 300, 120 |
| Number of algorithm iterations | 500 |
| Image width | 128px |
| Image Hight | 128px |
| Mode | Parallel |
| Algorithm | SART TV |
| TV lambda | 1000 |
| TViter | 1000 |

## Transmission  Scatter  Scatter + Transmission  LiDAR

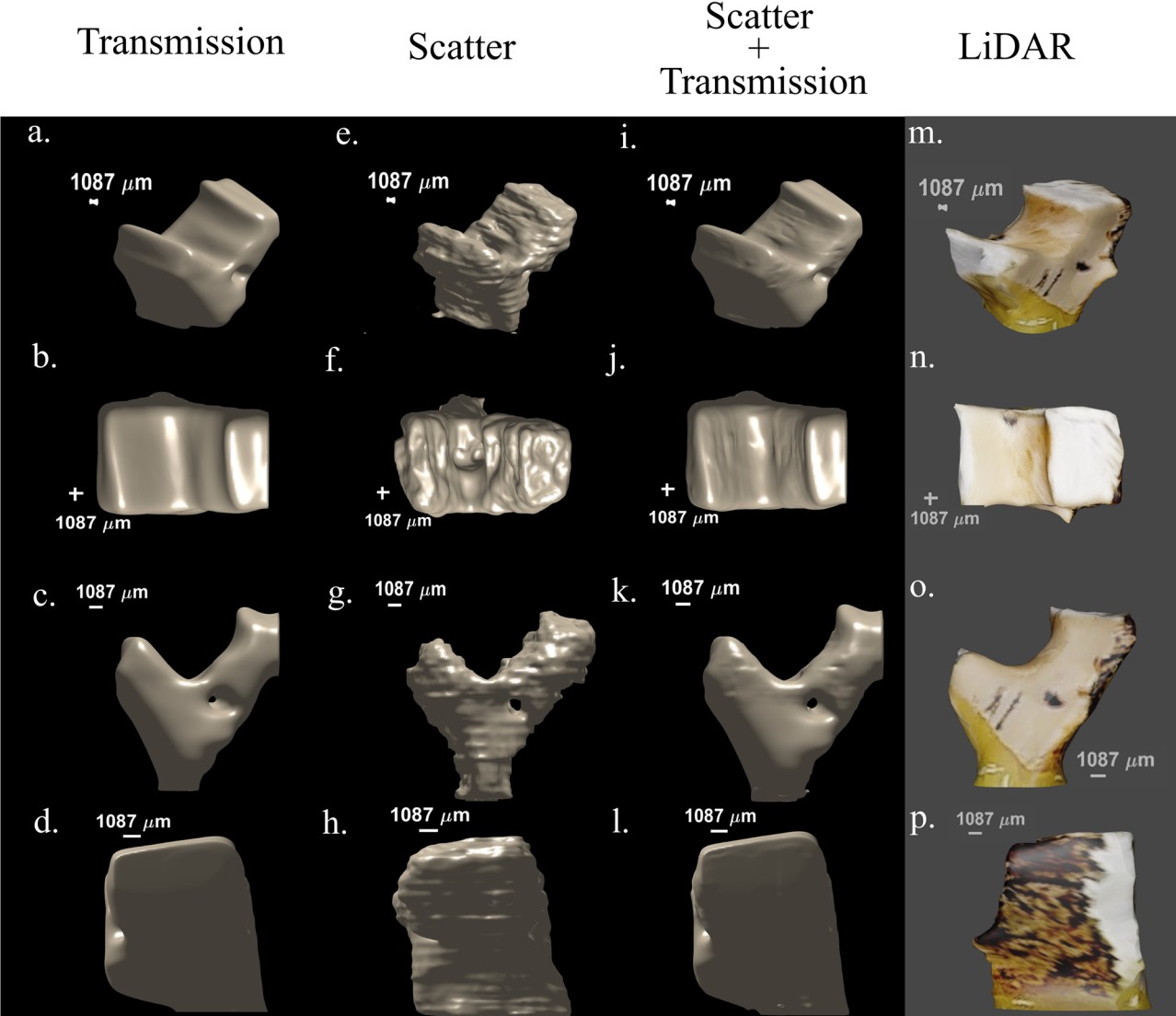

**Fig. 3 | Isosurface of the bone from various angles.** The isosurface was reconstructed using data obtained by different imaging modalities: (**a–d**) transmission, (**e–h**) scatter, and (**i–l**) combination of transmission and scattering. The transmission isosurface provides a general representation of the structure of the object while the scatter isosurface offers greater details thanks to its higher resolution and resilience to scattering noise. The combined isosurface shows the general structure of the object maintaining high resolution and displaying the transmission image with additional details. As a guide we added the corresponding 3D LiDAR (laser imaging, detection, and ranging) scans captured with iPhone pro (**m–p**).

correlation[50–52] of two different, 3000 samplings reconstuctions (see Supplementary Fig. 7). The cutoff frequency is ~0.001 $\mu m^{-1}$ and thus the spatial resolution is 500 $\mu m$ (Fig. 4l). We belive that the degredation in resolution is due to noise from our measuring system which was caused by ground loops in the setup.

Figures 3, 4 illustrate the potential advantages of our method, however, there are still several differences between the 3D reconstructions obtained by transmission and by scattering. These differences are highlighted in the tomograms of the bone taken from front to back, shown in Fig. 5. One noticeable difference is that with our method the object seems darker in its middle section. This is mostly due to self-absorption and self-scattering that led to a reduction of the signal measured by the detector. Although the differences result from the real physical differences between the mechanisms that govern absorption and scattering, they have to be understood or reconciled to provide the correct 3D structure of the object. This challenge can be mitigated by using another detector (or detectors) at a different angle to compensate for this effect (see Supplementary notes 3, 4 and Supplementary Figs. 3–5). We also note that in the tomogram reconstructed by the combination of scattering and transmission the self-attenuation is less pronounced suggesting that the modality can overcome the challenge with some improvement in the algorithms. A second prominent challenge is related to the tradeoff between the measurement time and the image quality. Since our method relies on scanning, the measurement time is proportional to the number of samplings taken. However, to achieve high image quality, even with clever reconstruction algorithms, the quality of the image is degraded when the number of samplings is much lower than the number of pixels.

Our work suggests that scatter radiation can be harnessed to create high resolution CT images, either on its own or in conjunction with traditional transmission information, resulting in a significant improvement in image quality. The selection of approach depends on the specific details of the application and constraints of the measurements. For instance, using scattering-only based CT allows the detector to be mounted at any angle, which can be advantageous in situations where access to certain locations is limited. By combining scatter and transmission information, it is likely that a smaller number of scanning points will be required, making this approach

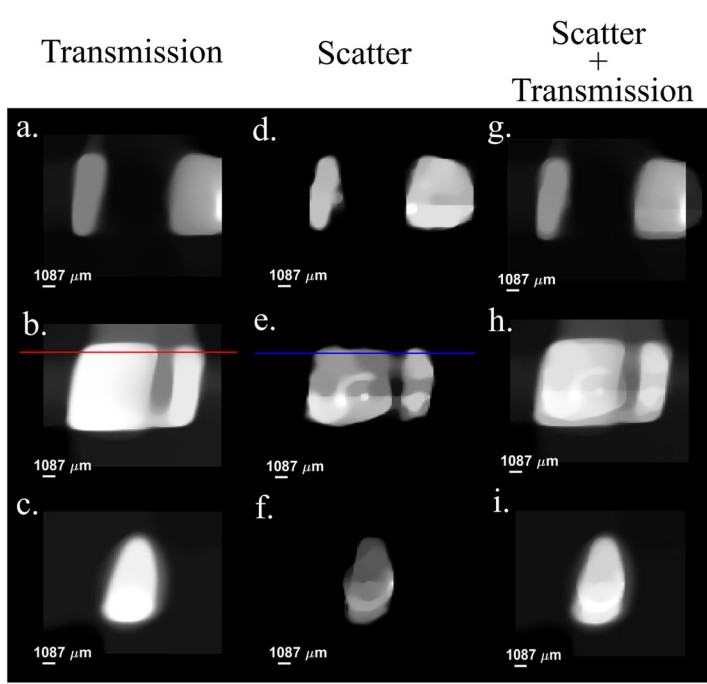

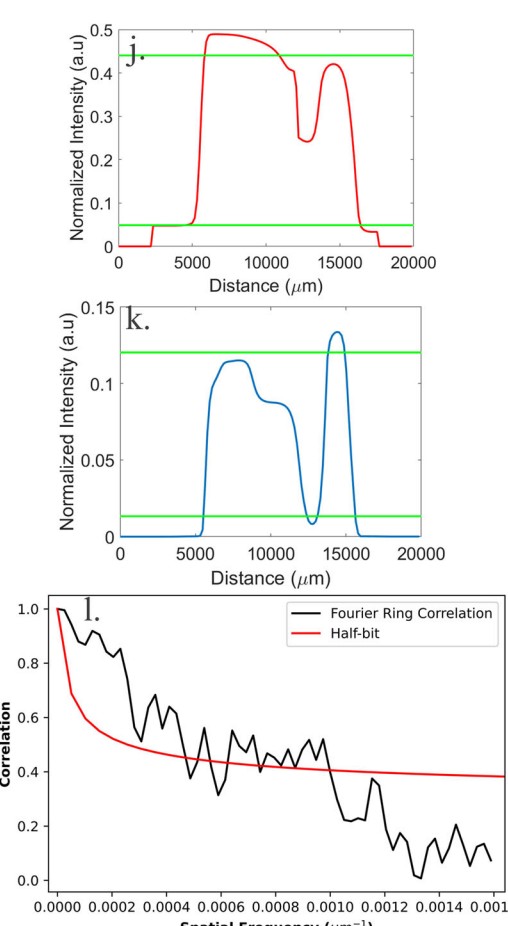

**Fig. 4 | Tomogram (cross-sectional) images of the bone.** Tomograms sliced from top to bottom: (**a**–**c**) transmission, (**d**–**f**) scatter and (**g**–**i**) a combination of both. The tomograms in each row correspond to the same slice that has been reconstructed using the respective modality. **j, k** Presents the cross sections of (**b**) and (**e**) respectively when the green lines represent 10% and 90% of the signal intensity in each graph. **l** Fourier ring correlation of the (**e**) tomogram as a function of the Fourier domain frequency. The cross between the FRC graph and the half-bit graph is approximately at 0.001 μm⁻¹, which indicates that the resolution of our system is at least 500 μm.

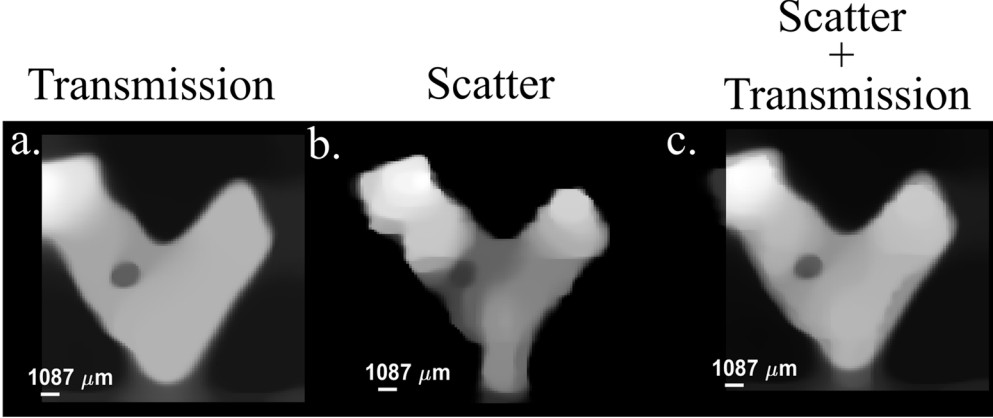

**Fig. 5 | Tomogram (cross-sectional) images of the bone.** Tomograms sliced from the front: (**a**) the transmission, (**b**) scatter, and (**c**) a combination of both.

more suitable when dose is a primary concern. Furthermore, our technique has the potential to enhance the image quality of medical CT images by offering higher capability to resolve small and complex details and lower radiation dose. This is made possible by the simultaneous collection of scattered photons from a 360-degree range. As a result, the need for grids that are currently used in medical X-ray measurements can be eliminated, simplifying the imaging process and further improving image quality. By discarding these grids, we can reduce image artifacts and improve the overall diagnostic accuracy of medical CT imaging.

## Methods
### Equipment details
The x-ray cone beam source in the experiment was a VJ Technologies P051, which was operated at a voltage of 80 kV and a current of 2 mA. To collimate

the beam, we used a 14 mm diameter, 500 mm long, 18 mm thick lead sheet, fashioned into a tube shape, with a slit mounted at its end. The silver mask was custom made using a special 3D printer by Nano Dimension Ltd[53] and it was mounted on two Thorlabs LTS150 stepper motors for precise positioning. We used a Thorlabs K10CR1 stepper motor rotation stage mounted on two Thorlabs MTS50-Z8 stepper motors to rotate the object and align it with respect to the beam. The detector of choice was a Saint Gobain Lanthanum Bromide scintillation detector paired with AS-20 photon multiplier tube. The detector is connected to the Amptek DP5G digital pulse processor and a Canberra 3102D high-voltage power supply. The flat panel detector used for capturing the transmission images of the object is the iRay Technology NDT0909M.

### Mask production and calibration

The mask was created using a patented unique process by Nano-dimension, which utilizes a multi-material multi-layer 3D printer that the company typically uses for fabricating electrical circuits with silver and polymer. We designed the mask to have $1480 \times 1480$ pixels with transmission-absorption ratio of 1:1 and dimensions of $160~\text{mm} \times 160~\text{mm} \times 1.5~\text{mm}$ with a feature size of 108 μm.

To circumvent the potential inhomogeneity in the absorption of the mask, we measured its transmission using a flat panel detector. The resolution of the images obtained from the transmission data, which we obtained by our flat panel detector is expected to be ~500 μm. We summed over all pixels in each measurement to normalize the data. To ensure accurate alignment with the manufacturing files on the computer, we used the same low-resolution flat panel detector to capture an image of the mask. We later established a correlation between this image and a low-resolution version of the manufacturing file, which enabled us to accurately locate the mask during the experiment.

### Deep neural network reconstruction tool parameters

Due to the absence of training data of x-ray scattering for high resolution images, we utilized self-supervised DNN algorithm for the reconstruction of the GI images. To accomplish this, we utilized the GIDC code, which provides high performance reconstruction but is only able to reconstruct images with number of pixels, that is a power of two in each axis. To meet this requirement, we resized our original image from $184 \times 163$ pixels to $128 \times 128$ pixels, resulting in a slight reduction in resolution. Also, GIDC was originally made for far-field super-resolution reconstructions in visible light. To adapt it to our use we changed the learning rate and the TV strength.

### Reconstruction computer specifications

We used an Intel core i7-10700 CPU combined with Nvidia Quadro 2200p GPU and 128 GB of RAM.

### Simulation details

Simulations were performed using the Monte-Carlo (MC) simulation code FLUKA 4-3.1[54,55] and the Flair 3.2–2[56] graphic user interface. We implemented our GI system within the code by modeling each laboratory component separately. The simulation consists of three elements: radiation source, object, and detectors (see Supplementary Fig. 1). For more technical information about the parameters of each element see Supplementary Table 1. A detailed method of implementing GI systems via MC simulations can be found in ref. [57].

For this simulation, 2000 realizations were performed, each consisting of 1 million primary particles, to a total of 2 billion particles. The simulation system consisted of three components:

- Radiation source—In the laboratory setup, the radiation field impacting the object is primarily defined by the X-ray tube and the mask. To achieve high resolution CGI we employed masks with thousands of fine features. However, representing these masks as physical objects in MC programs and changing them for each sampling is computationally demanding and time consuming. This is because

MC simulations track each primary photon separately and record the interactions along its path. To circumvent complexities associated with modeling thousands of masks and their interactions, including self-absorption, self-scatter, recoil, and energy loss, we adopted a more efficient approach for modeling the radiation source. In our simulation, we defined a radiation source that represents the field immediately after its interaction with the masks, positioned 5 cm before reaching the object. The 80 kVp energy spectrum was obtained with the SPEKTR 3.0 program[58] using the TASMICS algorithm. This spectrum was generated for a Fewell tube with a tungsten anode and inherent 1.6 mm Al filtration, without any additional filtration.

- Geometry simulation—We defined our object as a 0.85 cm radius sphere made of bone. The density and elemental composition of the bone were defined as the compact bone material from the Adult Reference Computational Phantoms presented in the ICRP publication 110[59]. The Imaging system was set in the air. Dimensions and material descriptions of each element in the imaging system are presented in Table 1.

- Detectors—Five detectors were defined, at $0^0$, $45^0$, $90^0$, $135^0$, and $270^0$ with regard to the imaging axis. Each detector was set as a $1.7 \cdot 1.7$ cm ideal boundary crossing detector. We used the USRBDX card in FLUKA that can count the particles crossing a boundary. We defined a one-way fluence detector counting the total of all passing photons.

### Sample materials

The bone sample is a small part of a cow bone that was bought at a butcher shop.

### Data availability

The data that support the findings of this study are available from the corresponding author upon reasonable request.

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

## Author contributions

Conceptualization: A.B., S.S., E.C. Methodology: A.B., O.S., Y.K. Investigation: A.B. Visualization: A.B., O.S. Monte-Carlo simulations: R.H.S. Funding acquisition: S.S., E.C. Project administration: H.S. Supervision: S.S. Writing—original draft: A.B. S.S. Writing—review & editing: A.B., O.S., Y.K., H.S., E.C., S.S.

## Competing interests

The authors declare no competing interests.
