## [Peer Review File · Communications Engineering]

Reviewers' comments:

Reviewer #1 (Remarks to the Author):

This paper reports a technique that measures scattered radiation in X-ray Computed Tomography using computational ghost imaging. In general, I think it is significant to find novel methods that combine scattered and transmission signals with improved image quality and opportunities for radiation dose reductions. That said, I have two major concerns about this manuscript: 1) In my opinion, the “main text” of the paper is hard to follow and comprehend. I think the content in the “main text” could be written with a more coherent flow, helping the readers to understand more easily. 2) My second concern is that most of the results are qualitative. I think the study can be more rigorous if more quantitative results are included, showing that the images of combined scatter and transmission produce higher image quality. My specific comments are detailed below.

Abstract:

- I think that abstract is written well. However, it does not provide quantitative and specific findings that support the claims.

Main text:

- Lines 55:59: Authors mentioned that they measured scattered X-ray radiation utilizing a structured illumination (SI) approach. The next sentence is about combining CGI and image reconstruction. It is hard to follow how the SI approach and CGI approach with reconstructions are connected.
- Line 66: Authors wrote “we apply it to incoherent scattered radiation”. Does that mean that you did not use the CGI approach to calculate the transmission signal? If so, how was the transmission signal calculated?
- I did not understand how the transmission signal is calculated. Further, I did not understand how the scatter and transmission signals are combined to for a higher quality image.
- There are various choices that are mentioned in the manuscript where authors did not justify their choices, e.g., 16384 pixels, 3468 iterations, 28 angles, or size of the beam mask (17 x 19.5), etc.
- Main text refers to Fig. 3-5 where Fig. 5 cannot be found.
- Fig. 3 and 4 are qualitative. I think it is important to demonstrate the improved resolution, contrast, and accuracy of density measurements in a quantitative manner.
- Fig. 4 demonstrate some topogram images. However, it is not clear if the image of combined transmission and scatter are able to quantify the density of the material more accurately.

Reviewer #2 (Remarks to the Author):

This paper mainly discusses that scattered photons in most traditional x-ray imaging will lead to blurring and a decrease in the resolution of imaging, so the x-ray computing ghost imaging scheme (computational ghost imaging, CGI) using scattered radiation is proposed to improve the imaging quality

and resolution under the premise of reducing the radiation dose. This paper proposed to 1) produce 3D printed silver binary mask, which modulates cone x-ray beam to produce structured illumination beam incident on the object and 2) collect the total intensity signal scattered from the object in the direction of orthogonal to the beam. The projection image of each angle is reconstructed by deep neural network combined with the regularization of total variation, and the CT slices are reconstructed by algorithm of SART-TV to achieve higher quality and resolution in the premise of missing projection angle. Comments are as follows :

1. Results of the context is inadequate, and in addition, lack of sufficient analysis.
2. What is the approximate X-ray dose of this experiment? How much lower is it as compared to conventional methods?
3. If we expect the proposed method has comparable level of CT image quality in regular medical scans, how much dose (and time cost) might be necessary?
4. Is there a physical explanation to replace the label of “arbitrary units” in the vertical coordinate of Fig. 4 (j-k)? Such as relative intensity/density? What does the green lines in these plots mean?
5. Page 1 of 16 line 23. ‘We show that our method can provide sub-200 μ m resolution, exceeding the capabilities of most existing X-ray imaging modalities.’ It is more accurate to describe it as “exceeding the capabilities of most existing X-ray medical imaging modalities”.
6. Page 1 of 16 line 34. The authors mentioned ‘In medical imaging, for instance, denser structures such as bones absorb more X-rays and appear white on the image, while softer tissues, like muscles and inner organs, absorb fewer X-rays and appear darker.’ However, in Page 12 of 16, line 405, we find the description of ‘with areas containing more absorbing material appearing darker’, it is clearly contradictory.
7. Page 3 of 16 line 68. ‘It is worth mentioning that although cameras based on the Compton effect are available (34, 35), their resolution is poor due to the tendency of scattered radiation to blur. As a result, they are not suitable for medical imaging, unlike our proposed method which provides higher resolution and better image quality (35, 36).’ Style of reference citing errors.
8. Authors seem to use the term “iterations” to imply the number of the measurement times or the length of the signal S (as indicated in line 99 and 173-176, in accordance with the sampling rate offered in supplementary materials: $SR=0.211=3468/128/128$). However, in cases of deep-learning, “iterations” usually refers to the number of training epochs or the optimization steps, the same as it is in GIDC work. It’s better to substitute “iterations” with “measurements” or “number of sampling” to avoid unnecessary misunderstanding.
9. Page 3 of 16 line 91. Three questions concerning $T_{\phi^*} = \left[\operatorname{argmin}_{\phi} |AT_{\phi}(x_{DGI}) - S|^2 + \tau\mu[T_{\phi}(x_{DGI})] \right]$ of Eq.(2) (1) x_{DGI} , what the subscript of DGI denotes? Differential ghost imaging? If so, how was DGI applied in the algorithm? (2) One dimensional sequence signal $S = T_{\phi}(x_{DGI})$ should be close to the primary one dimensional signal S. This means that $|AT_{\phi}(x_{DGI}) - S|^2 = |AS - S|^2$ may become larger. What does it mean? ; (3) τ, μ represent the full variational coefficient and the regularization parameter, respectively, where is the difference between these two parameters?
10. Page 3 of 16 line 102. ‘To achieve this, we repeated the procedure multiple times while rotating the object, ultimately reconstructing the 3D image from 28 different angles.’ 28 projections are quite limited and interior image quality can be achieved. Is it possible to reconstruct the CT slices with more projections? If impossible, what are the main obstacles?
11. Page 4 of 16 line 105. ‘a 500 mm collimator (not shown) that is mounted between the source and the

object to minimize scattering from the surrounding environment'. How much does the divergence of X-rays increase after the collimator? Besides, after collimation tomogram slices are still reconstructed dependent on conical beam, why?

12. Page 4 of 16 line 124. In Eq. (3) , what does the minute differential cross section mean? Does it benefit or harm the experiments?

13. Page 6 of 16 line 138. 'It is evident that the scatter reconstruction reveals finer features on the surface of the object compared to the transmission reconstruction, indicating that our method has a higher resolution.' Structure revealed from the scatter reconstruction might be fluctuations or noises rather than fine details, which is hard to be confirmed by LiDAR images alone. Is there any evidence that demonstrate that scattering reconstruction can achieve higher resolution?

14. Page 6 of 16 line 140. 'The resolution of the images obtained from the transmission data is expected to be approximately 0.5 mm, which is the limit of our detector'. Why is the image resolution limited by the detector, other than FWHM of autocorrelation function of the mask?

15. Page 7 of 16 line 174. 'Since our method relies on scanning, the measurement time is proportional to the number of iterations.' More details about the scanning should be supplied for easy understanding how to reduce the measurement time.

16. Page 10 of 16 line 313. 'We designed the mask to have 1480x1480 pixels with transmission-absorption ratio of 1:1, and dimensions of 160mm x 160mm x 1.5mm with a feature size of 108 μ m.' The aperture size of the mask is 108 microns, but the FWHM of the autocorrelation function in the horizontal and vertical directions is 175 and 127 microns respectively. Why the conical beam amplification in the horizontal and vertical directions is different?

17. Figure legend for Extended Data Fig.4 is hard to be understood.

Reviewer #3 (Remarks to the Author):

This paper reports a very important result, namely the use of penetrating scattered radiation to demonstrate genuinely volumetric imaging (i.e. tomography rather than merely topography) using a computational ghost-imaging strategy. The point regarding the useful but very-often discarded/wasted information in scattered x-ray radiation (in the context of x-ray imaging) is powerful, convincing and significant. Again, this paper is clearly a very important contribution, both to the field of x-ray ghost imaging in particular, and to tomographic imaging more generally. I very strongly recommend publication, provided that the points raised below are properly taken into consideration in preparing a revised version of the paper. If required by the Editor, I would be very happy to review such a revised form of the paper.

1. The basic geometry, in Fig. 2 of the present paper, may be compared to the proposal in Fig. 5b of A. M. Kingston et al., "Neutron ghost imaging", *Physical Review A* 101, 053844 (2020). While the latter paper considers neutrons rather than x-ray radiation as the penetrating probe, the key concept appears to be similar, namely that scattered radiation may be used to perform tomography via computational ghost imaging employing penetrating radiation. This paper by Kingston et al. should therefore be cited, and very briefly commented upon in the revised manuscript.

2. Line 57: When first mentioning computational ghost imaging, it would be useful for many readers to

give a suitable reference, e.g. to J. H. Shapiro, "Computational ghost imaging", Phys. Rev. A 78, 061802(R) (2008). This would be of benefit to readers who are not already familiar with the concept of computational ghost imaging.

3. Line 66, references 30-33 are cited for x-ray ghost imaging using transmitted light. Reference 30 (published in 2016) should be augmented by a second relevant paper published in the same year, namely H. Yu et al., "Fourier-transform ghost imaging with hard X rays," Phys. Rev. Lett., vol. 117, no. 11, 2016, Art. no. 113901. Adding this citation would then mean that both of the original 2016 x-ray ghost-imaging papers were cited.

4. Figure 3 would benefit from a spatial scale bar, similar to that which is given in Fig. 4. Also, regarding Fig. 3, is there some way of more strongly indicating that the finer isosurface features in column 2 (relative to the smoother isosurfaces in column 1) are indeed genuine fine-detail information rather than artefact, beyond the indications that are currently given in the main text?

5. How was the tomogram combination performed, e.g. in passing from columns 1 and 2 to column 3 in Fig. 4? Just a little more detail, on this point, would be helpful for the reader's understanding of the paper. Also, why are four significant figures needed for the spatial-scale-bar label, in Fig. 4?

6. Lines 151-154 state that "unlike previous works on 3D reconstructions using single-pixel detectors with visible light [41,42], which recovered only the surface gradients to derive the 3D surface of the object, we reconstructed a 3D volume that contains information about both the internal parts and the surface of the object". In this context, the following paper should certainly be cited and briefly commented upon, since it employed computational ghost imaging to perform genuinely 3D reconstructions, albeit with transmitted rather than scattered x-ray radiation: A.M. Kingston et al., "Ghost tomography", Optica 5, 1516-1520 (2018).

Minor comments:

7. Line 15: Perhaps "opaque instruments" should be something more like "other opaque samples"?

8. Lines 68-71: The reference style changes from numbers in superscript font, to numbers in non-superscript font in round brackets. Also, are the cited references correct, in this paragraph?

9. Lines 173-176 state that "Since our method relies on scanning, the measurement time is proportional to the number of iterations. However, to achieve high image quality, even with clever reconstruction algorithms, the quality of the image is degraded when the number of iterations is much lower than the number of pixels". Is this the same use of the term "iteration", in the sense of "iterations of an image-processing code", that is given in line 89? I ask this because the words "the measurement time is proportional to the number of iterations" might be read (or misread?) as stating that the experimental measurement time is directly proportional to the number of computational iterations required to analyse the data? Please clarify the text here.

10. Line 310 speaks of a "unique process". Why is the indicated process unique?

11. Line 316 speaks of a “low-resolution flat panel detector”. What is the resolution for this particular detector?
12. Line 330, “Far-field” should not begin with a capital letter.
13. Line 334, “#learning rate” should not be indented differently to the other comments.
14. Line 341, “intel” should be “Intel”.
15. Line 353, a little more specific detail regarding the nature of the “computational considerations” would be useful for the reader.
16. Line 398, should “is shown” be “, as shown”?
17. Line 402, is the word “in” missing after “function”?
18. Line 422, “could” should probably be “would”.
19. Extended data, Fig. 2, “determanation” is incorrectly spelled. Extended data, Fig. 3, “experimentl” is incorrectly spelled. Similarly for Extended data, Fig. 4, whose caption has four spelling errors.
20. The references need to be properly proofread. For example, weblinks appear in place of the journal name in references 2 and 4, the journal name in reference 5 is not abbreviated, reference 31 has an asterisk that should not be there, reference 44 is missing its page number, reference 47 is missing journal details and page number, reference 50 is missing its article number, etc.

High-resolution computed tomography with scattered X-ray radiation and a single pixel detector

A. Ben Yehuda¹, O. Sefi¹, Y. Klein¹, R. H Shukrun^{1,3}, H. Schwartz¹, E. Cohen², and S. Shwartz^{1*}

Detailed response to the comments of the Reviewers

A detailed response to the Reviewer 1

Reviewer 1 has two main concerns: one regarding the clarity of the main text, and the other related to the presentation of quantitative results. We appreciate the valuable feedback from the reviewer and have made the necessary revisions to the manuscript, as detailed below.

Reviewer comment (1):

- Authors mentioned that they measured scattered X-ray radiation utilizing a structured illumination (SI) approach. The next sentence is about combining CGI and image reconstruction. It is hard to follow how the SI approach and CGI approach with reconstructions are connected.

Our reply to comment (1):

We thank the reviewer for bringing this to our attention. Indeed, we were not precise with the terminology we used. When mentioning structured illumination, we were referring to the intensity pattern introduced by the mask we used for the CGI. To make the text clearer we have revised the phrase structured illumination to ghost imaging approach. We have also revised the phrase “SI” to “patterns”.

Reviewer comment (2):

- Authors wrote “we apply it to incoherent scattered radiation”. Does that mean that you did not use the CGI approach to calculate the transmission signal? If so, how was the transmission signal calculated?

Our reply to comment (2):

We apologize for not being sufficiently clear. When we mentioned “applying it to incoherent scattered radiation” we were referring to the measurement of Compton scattering at the bucket detector, which is a type of incoherent scattering. It is important to note that due to the close proximity of the mask to the detector, propagation effects were negligible. As a result, we utilized the fabricated patterns as our reference data. To enhance the clarity of the sentence we have revised it to “While X-ray CGI has been previously demonstrated with single-pixel detectors for collecting transmitted³⁵⁻³⁹, fluorescent⁴⁰, and refracted radiation⁴¹, we apply this technique to

incoherent (Compton) scattered X-ray radiation. This approach is akin to the method proposed by A. M Kingston et al. for imaging of scattered neutrons⁴².” (page 3 first paragraph)

Reviewer comment (3):

- I did not understand how the transmission signal is calculated. Further, I did not understand how the scatter and transmission signals are combined to for a higher quality image.

Our reply to comment (3):

We appreciate the comment. To calculate the transmission signal, we directly measured the image of the object using a standard X-ray detector. We then combined it with the images we obtained using our Compton CGI method. To combine both images we averaged the negative of the transmission image with the reconstructed scatter image. The key idea is that our method can provide information on the object is obscured by scattering, which is known to introduce blurring in standard X-ray imaging, see for example R.B. Wilsey, “Scattered X-rays in x-ray photography”, in *Journal of the Franklin Institute*, vol. 194, Issue 5, pp. 583-596, 1922, In contrast, our method like other GI methods, as has been shown in many publications such as Q. Chen, S. K. Chamoli, P. Yin, X. Wang and X. Xu, "Active Mode Single Pixel Imaging in the Highly Turbid Water Environment Using Compressive Sensing," in *IEEE Access*, vol. 7, pp. 159390-159401, 2019, is more robust against scattered radiation, hence can provide finer details. However, as with any GI method, the image quality with our method is highly dependent on the several factors which can limit the performance including the number of measurements, noise levels, and the precision of the mask. In this work we highlight the potential benefits of combining our method with the standard approach. Importantly, this combination can be implemented simultaneously without increasing the radiation dose, potentially leading to improved results in terms of resolution, image quality, and radiation dose efficiency. To enhance clarity, we have included additional explanatory text in the manuscript:

“To perform a 3D image reconstruction using tomography techniques, it is essential to obtain projections of the object from various angles. We successfully accomplish this for both transmission and scatter images. In the case of transmission, we captured the images directly with a flat panel detector. For the scatter images we employed our CGI technique. We repeated the procedure multiple times, rotating the object each time, which allowed us to reconstruct 3D images from 28 different angles.” (Page 4 second paragraph)

“To validate our method and to benchmark its efficiency, resolution, and sensitivity against the standard direct transmission CT reconstruction, we reconstructed 3D images of a bone using signals obtained from transmission negative images, scattering reconstruction images, and their average. The results of these reconstructions are presented in Figs. 3-4” (page 6, second paragraph).

Reviewer comment (4):

- There are various choices that are mentioned in the manuscript where authors did not justify their choices, e.g., 16384 pixels, 3468 iterations, 28 angles, or size of the beam mask (17 x 19.5), etc.

Our reply to comment (4):

This is a very good point, and we agree that it is important to provide the requested information. The selection of parameter values was a choice aimed at achieving high image quality while minimizing the number of measurements, all within the constraints of our system. For example, we found that 3468 iterations were sufficient to obtain the best image quality our system and reconstruction process could deliver. Additional measurements beyond this point did not yield a significant improvement in image quality. To illustrate this point, we have added an explanation, along with a convergence factor calculation and a corresponding graph in the Supplementary Materials (page 14 paragraph 3). Regarding the choice of 16384 pixels, this selection is directly related to factors such as the beam size, the mask feature size, and the binning necessary for data adjustment within the framework of the GIDC algorithm that operates 128x128 pixels. We chose 28 angles because this number yielded satisfactory results that demonstrate the proof of principle of our method.

Reviewer comment (5):

- Main text refers to Fig. 3-5 where Fig. 5 cannot be found.

Our reply to comment (5):

The reviewer is correct. We have revised the reference to: "Fig. 3-4".

Reviewer comment (6):

- Fig. 3 and 4 are qualitative. I think it is important to demonstrate the improved resolution, contrast, and accuracy of density measurements in a quantitative manner.

Our reply to comment (6):

We greatly appreciate the reviewer's valuable input and fully acknowledge the importance of incorporating quantitative plots to enhance the quality of our manuscript. To provide a quantitative assessment of the resolution, contrast, and accuracy of our density measurements we have added a plot of the Fourier ring correlation (FRC) for our reconstructed images. The FRC indicates that the resolution of the reconstruction is about 0.5 mm. We have plotted the results in Fig. 4o and added to the following text in page 7 paragraph 2: "To further explore the ability of our method to resolve fine details, we evaluated the reconstruction resolution by measuring the cutoff frequency of the Fourier ring correlation (FRC) of two different, 3000 samplings

reconstructions (Extended Data Fig.7) .The cutoff frequency is approximately $0.001 \mu\text{m}^{-1}$ and thus the spatial resolution is $500 \mu\text{m}$ (Fig. 4o).”

To provide a quantitative assessment of the contrast we have added an example with the cross section as indicated by the red and blue lines, respectively in Fig 4.j and Fig 4.k. The cross sections of the tomograms indicates that contrast of the scatter image is higher compared to the transmission image. We have added an explanation on page 7 paragraph 2: “This enables us to present tomogram slices from top to bottom of the object as is shown in Fig. 4a to Fig. 4i. We specifically focused on a small hole in the bone, whose size is approximately $2000 \mu\text{m}$ on one end and $1200 \mu\text{m}$ on the other and presented its cross sections in Fig. 4j and Fig. 4k. The images and cross sections reveal that the edge of the hole is nearly imperceptible in the transmission tomogram due to scattering. In contrast, the scatter tomogram remains resilient to this effect and clearly depicts the hole. This indicates that our method provides higher resolving power compared to standard transmission-based CT.”

Unfortunately, due to the distinct underlying mechanisms of scattering and absorption, the scatter images differ from the transmission images. Consequently, direct comparisons between them are not possible. Additionally, as we used a natural bone, we lack a ground truth object for comparison.

Reviewer comment (7):

Fig. 4 demonstrate some topogram images. However, it is not clear if the image of combined transmission and scatter are able to quantify the density of the material more accurately.

Our reply to comment (7):

We would like to thank the reviewer for pointing out this concern. Our manuscript constitutes a proof-of-concept study, and thus, we emphasize the significance of highlighting various possibilities for enhancing the performance of x-ray imaging modalities using our method. The presentation of the combined transmission and scatter image suggests a potential route for achieving higher-contrast, higher-resolution imaging. In principle, the combined transmission and scatter images can reveal fine details that are blurred in the transmission reconstruction due to scattering. This is attributed to the greater resilience of our method (GI) to scatter blur compared to standard transmission-based CT. We have added the following explanation to the manuscript: “The combined reconstruction reveals features which are blurred in the transmission reconstruction due to scattering. This suggests a new approach to eliminate the need for collimators after the object while maintaining high-quality images despite scattering. Adopting this approach has the potential to significantly reduce radiation exposure as collimators absorb a substantial amount of radiation, which represents lost information that could have been collected as demonstrated in our experiment.” (page 7 paragraph 1).

Summary: We would like to express our gratitude to the reviewer for the meticulous review of our manuscript and the valuable comments provided, which have significantly enhanced the clarity and strength of our arguments. We believe that we have effectively addressed all the reviewer's concerns and kindly request their renewed consideration of our manuscript for publication in Communications Engineering in its current form.

A detailed response to the Reviewer 2

We thank Reviewer 2 for the valuable assessment of our manuscript. In response to the reviewer's insightful comments, we have addressed all of them as outlined below:

Reviewer comment (1):

- Results of the context is inadequate, and in addition, lack of sufficient analysis.

Our reply to comment (1):

We greatly appreciate the reviewer for bringing this crucial aspect to our attention, and we agree that our manuscript would benefit from additional analysis and the inclusion of quantitative data. To address this, we have incorporated several additional quantitative plots into the manuscript, including:

Fig. 4o - Fourier ring correlation of the (e) tomogram as a function of the Fourier domain frequency. We explain how we use this technic to quantify the resolution we achieve in the manuscript: “by measuring the cutoff frequency of the Fourier ring correlation (FRC) of two different, 3000 samplings reconstructions (Extended Data Fig.7) .The cutoff frequency is approximately $0.001 \mu\text{m}^{-1}$ and thus the spatial resolution is $500 \mu\text{m}$ (Fig. 4o). We believe that the degradation in resolution is due to noise from our measuring system which was caused by ground loops in the setup.”(page 7 paragraph 2)

Extended Data Fig.6 – To demonstrate the relationship between image quality and the number of samplings we introduced the Convergence Factor defined as - $C_F = \frac{1}{N \cdot M} \sum_{i,j=1}^{N,M} (x_{i,j} - b_{i,j})^2$. We have added a figure that illustrates the dependence of the convergence factor on the number of samplings on page 14. The data in the figure indicate that the convergence factor does not change significantly beyond 3000 samplings, hence additional sampling is unlikely to significantly improve the quality of reconstructions. We have addressed the following in the manuscript: “We chose this number of samplings because we observed negligible improvement of the image quality when the number of samplings exceeded this value (see Supplementary Materials).” (page 3 last paragraph)

Reviewer comment (2&3):

- What is the approximate X-ray dose of this experiment? How much lower is it as compared to conventional methods?

- If we expect the proposed method has comparable level of CT image quality in regular medical scans, how much dose (and time cost) might be necessary?

Our reply to comment (2&3):

We appreciate the reviewer's valuable feedback. The primary objective of our experiment was to demonstrate the viability of using X-ray ghost imaging as a method for reconstructing images from scattered radiation, which we believe is a significant step forward in this field. As far as we are aware, this experiment marked the initial attempt to establish this feasibility, and thus our primary focus was on achieving this goal. To achieve the goal, we had to overcome numerous challenges including blocking unwanted noise and unknown scatter, producing masks which can modulate high energy X-ray at a high resolution, reconstruct an image with limited number of samplings etc. Unfortunately, we were unable to directly measure the dose, as our detector is highly sensitive and unsuitable for direct insertion into the X-ray beam, and it is not dose calibrated.

We agree with the reviewer that assessing the dose is an important aspect of radiation experiments. However, we'd like to clarify that our experiment did not aim to optimize the system or image reconstruction software. As such, estimating the dose, which could be a comprehensive study in itself, was not within the scope of our manuscript. We acknowledge the significance of the points raised by the reviewer and agree that they should be addressed in future research. Your feedback is greatly appreciated and will be considered for future work.

Reviewer comment (4):

- Is there a physical explanation to replace the label of "arbitrary units" in the vertical coordinate of Fig. 4 (j-k)? Such as relative intensity/density? What does the green lines in these plots mean?

Our reply to comment (4):

The green lines represent 10% and 90% of the signal intensity in each graph to emphasize that our method is capable of mitigating blurring due to scattering, which is a great challenge in standard transmission CT modalities. Accordingly, we have added an explanation to Fig. 4. Info: (j) and (k) presents the cross sections of (b) and (e) respectively when the green lines represent 10% and 90% of the signal intensity in each graph". We used "arbitrary units" because our detector is not dose calibrated. Nevertheless, the counts registered by the detector are directly proportional to the scattered radiation, making their spatial distribution indicative of variations in electron density within the sample. While we acknowledge that precise detector calibration is essential, it typically necessitates the availability of a calibrated source or detector. Unfortunately, we currently lack access to such calibrated equipment.

We have added this explanation to the manuscript "The images and cross sections reveal that the edge of the hole is nearly imperceptible in the transmission tomogram due to scattering. In contrast the scatter tomogram remains resilient to this effect and clearly depicts the hole. This indicates that our method provides higher resolving power compared to standard transmission-based CT."(Page 7 paragraph 2)

Reviewer comment (5):

“We show that our method can provide sub-200 μm resolution, exceeding the capabilities of most existing X-ray imaging modalities.’ It is more accurate to describe it as “exceeding the capabilities of most existing X-ray medical imaging modalities”.

Our reply to comment (5):

After carefully considering this valuable feedback, we conducted a thorough assessment of our method's resolution using Fourier Ring Correlation. This analysis takes into account both the theoretical resolution and the impact of noise, providing a more comprehensive evaluation. The results of this updated analysis reveal a resolution of 500 μm . It's worth noting that this resolution is slightly lower than what was initially predicted by the autocorrelation method due to the presence of noise. Nevertheless, we have decided to cite the more conservative assessment of the resolution we present in the manuscript and revised it to 500 μm . However, we want to emphasize that the resolution achieved in our work represents a significant advancement, approximately an order of magnitude better than conventional X-ray imaging modalities based on scattered radiation. It's important to acknowledge that the noise observed in our system is a result of its imperfections, and we are actively exploring ways to enhance our method's performance. The detailed analysis and discussion on this topic have been included in the manuscript. We have revised the following sentence in the abstract: “We show that the resolution of our method can exceed 500 μm , which is approximately an order of magnitude higher than the resolution existing X-ray imaging modalities based on scattered radiation.”

Reviewer comment (6):

Page 1 of 16 line 34. The authors mentioned 'In medical imaging, for instance, denser structures such as bones absorb more X-rays and appear white on the image, while softer tissues, like muscles and inner organs, absorb fewer X-rays and appear darker.' However, in Page 12 of 16, line 405, we find the description of 'with areas containing more absorbing material appearing darker', it is clearly contradictory.

Our reply to comment (6):

We appreciate the reviewer for bringing up this concern and apologize for any confusion caused by our manuscript. We aimed to describe two different scenarios: the first involves absorption by the object when the source of radiation is the X-ray source, while the latter is an effect of self-absorption of scattered X-ray when the object itself acts as the source of measured radiation and as an absorber. To address this, we have made the following revisions to the manuscript to provide further clarification on this distinction.” As shown in Extended Data Fig.4 each detector produced a slightly different image reconstruction, with areas containing more absorbing material appearing darker. This is due to self-absorption of the scattering object.” (page 13 last paragraph).

Reviewer comment (7):

Page 3 of 16 line 68. 'It is worth mentioning that although cameras based on the Compton effect are available (34, 35), their resolution is poor due to the tendency of scattered radiation to blur. As a result, they are not suitable for medical imaging, unlike our proposed method which provides higher resolution and better image quality (35, 36).' Style of reference citing errors.

Our reply to comment (7):

We thank the reviewer for pointing out the error, and we have made the necessary correction.

Reviewer comment (8):

Authors seem to use the term “iterations” to imply the number of the measurement times or the length of the signal S (as indicated in line 99 and 173-176, in accordance with the sampling rate offered in supplementary materials: $SR=0.211=3468/128/128$). However, in cases of deep-learning, “iterations” usually refers to the number of training epochs or the optimization steps, the same as it is in GIDC work. It's better to substitute “iterations” with “measurements” or “number of sampling” to avoid unnecessary misunderstanding.

Our reply to comment (8):

This is a very good suggestion made by the reviewer. We have revised the manuscript accordingly and replace "iterations" with "samplings".

Reviewer comment (9):

Page 3 of 16 line 91. Three questions concerning $T_{\varphi}(\varphi^*) = \underset{\varphi}{\text{argmin}} |AT_{\varphi}(x_{\text{DGI}}) - S|^2 + \tau\mu[T_{\varphi}(x_{\text{DGI}})]$ of Eq.(2) (1) x_{DGI} , what the subscript of DGI denotes? Differential ghost imaging? If so, how was DGI applied in the algorithm? (2) One dimensional sequence signal $S_{\varphi} = T_{\varphi}(x_{\text{DGI}})$ should be close to the primary one dimensional signal S. This means that $|AT_{\varphi}(x_{\text{DGI}}) - S|^2 = |AS_{\varphi} - S|^2$ may become larger. What does it mean? ; (3) τ, μ represent the full variational coefficient and the regularization parameter, respectively, where is the difference between these two parameters?

Our reply to comment (9):

(1): The notation was used erroneously. DGI was not applied in our algorithm. Therefore, we have revised “ x_{DGI} ” to “ x ”.

(2) Regrettably, we encountered some difficulty in comprehending the question. If we understand the question correctly, it is true that we had a typo 3 lines below Eq. 2 which we have corrected, hopefully in accordance with the reviewer's comment, i.e. we changed the term $\tilde{S} = T_{\varphi}(x)$ to $\tilde{S} = AT_{\varphi}(x)$ in page 3, last paragraph.

(3): To make it clearer for the reader we have revised the corresponding paragraph. to reduce potential confusion, We have amended the term "full variational coefficient" to the term "TV operator". \mathfrak{I} represents the TV operator that works on the reconstructed image and τ denotes the regularization parameter, which allows us to determine the degree of sparsity to enforce on the minimized term." (page 3 last paragraph)

Reviewer comment (10):

Page 3 of 16 line 102. 'To achieve this, we repeated the procedure multiple times while rotating the object, ultimately reconstructing the 3D image from 28 different angles.' 28 projections are quite limited and interior image quality can be achieved. Is it possible to reconstruct the CT slices with more projections? If impossible, what are the main obstacles?

Our reply to comment (10):

These are very important questions. While using additional projections is possible, it would significantly increase the time required. There are no fundamental challenges; it is just that using more projections would necessitate a higher dose, which is typical in any CT system. Our goal in the present work was to demonstrate the feasibility of using scattered radiation. We have added an explanation to the manuscript "We repeated the procedure multiple times, rotating the object each time, which allowed us to reconstruct 3D images from 28 different angles. This number of angles proved sufficient for our relatively simple object. For more complex objects, it is likely that more angles will be required, but this can be achieved as in any standard CT scan." (page 4 paragraph 2)

Reviewer comment (11):

Page 4 of 16 line 105. 'a 500 mm collimator (not shown) that is mounted between the source and the object to minimize scattering from the surrounding environment'. How much does the divergence of X-rays increase after the collimator? Besides, after collimation tomogram slices are still reconstructed dependent on conical beam, why?

Our reply to comment (11):

We thank the reviewer for bringing to our attention that this point in the manuscript was not clear enough. We have added an explanation: "We implemented the GI with our scattered radiation scheme using the experimental setup illustrated in Fig. 2. It includes an X-ray source with parameters tuned to 80 kVp and 2 mA. To minimize scattering from the surrounding environment we employed 500 mm long circular collimator with a radius of 7.5 mm (not shown), positioned between the source and the object. The beam divergence after passing through the collimator was estimated to be 0.85 degrees. We used a slit (not shown) to reduce the beam size at the object to

$17 \cdot 19.5 \text{ mm}^2$, which is comparable to the size of the object. The spatial modulation of the beam immediately before reaching the object was achieved by a mask comprised of absorbing silver features. These features have transverse dimensions of roughly $100 \text{ }\mu\text{m}$ and thickness of about $1500 \text{ }\mu\text{m}$. The distance between the source and the mask is 1300 mm and the object is located 50 mm downstream of the mask. The object underwent 360° rotation during the measurements, facilitated by a rotation stage (not shown).”(page 4 paragraph 3)

Reviewer comment (12):

Page 4 of 16 line 124. In Eq. (3) , what does the minute differential cross section mean? Does it benefit or harm the experiments?

Our reply to comment (12):

We employ the minute differential cross section to show that even when we measure at an angle where the cross section is small, we can still reconstruct high quality images. To improve the clarity of the manuscript we have revised it as follows: “To demonstrate the integrity of our imaging method, the detector was mounted at approximately 90° relative to the input beam since the differential cross section given by the Klein-Nishina formula at this angle is small⁴² (low scattering angle)” (page 5 second paragraph).

Reviewer comment (13):

Page 6 of 16 line 138. ‘It is evident that the scatter reconstruction reveals finer features on the surface of the object compared to the transmission reconstruction, indicating that our method has a higher resolution.’ Structure revealed from the scatter reconstruction might be fluctuations or noises rather than fine details, which is hard to be confirmed by LiDAR images alone. Is there any evidence that demonstrate that scattering reconstruction can achieve higher resolution?

Our reply to comment (13):

We thank the reviewer for this comment. To address this concern, we have added a quantitative plot of Fourier Ring Correlation analysis and a discussion to the manuscript as elaborated in our reply to comment (1). We have incorporated quantitative plot into the manuscript Fig. 4o - Fourier ring correlation.

Reviewer comment (14):

Page 6 of 16 line 140. ‘The resolution of the images obtained from the transmission data is expected to be approximately 0.5 mm , which is the limit of our detector’. Why is the image resolution limited by the detector, other than FWHM of autocorrelation function of the mask?

Our reply to comment (14):

We apologize for not being clear in the previous version of manuscript. The Transmission data are just X-ray images and not CGI. To address this concern, we have added an explanation to the text: “The spatial resolution of images obtained by direct imaging with a flat panel detector is determined by the resolution of the detector, which we estimate to be about 500 μm , reflecting the blurring by the scintillation screen.” (page 7 first paragraph)

Reviewer comment (15):

Page 7 of 16 line 174. ‘Since our method relies on scanning, the measurement time is proportional to the number of iterations.’ More details about the scanning should be supplied for easy understanding how to reduce the measurement time.

Our reply to comment (15):

We appreciate the comment and agree that further information should be provided. We have added further details about the scanning process on page 12:

Scanning procedure

The measurement time at each mask position was 5 seconds. We repeated measurements for 3468 different mask positions at each angle of the sample, totaling 28 object angles. The angle of the object was adjusted using a rotation stage after every mask scan.

Reviewer comment (16):

Page 10 of 16 line 313. ‘We designed the mask to have 1480x1480 pixels with transmission-absorption ratio of 1:1, and dimensions of 160mm x 160mm x 1.5mm with a feature size of 108 μm .’ The aperture size of the mask is 108 microns, but the FWHM of the autocorrelation function in the horizontal and vertical directions is 175 and 127 microns respectively. Why the conical beam amplification in the horizontal and vertical directions is different?

Our reply to comment (16):

We apologize for any confusion. It is important to clarify that we conducted two distinct experiments using two different systems. We acknowledge that our manuscript may not have conveyed this point as clearly as intended. The sentence: “We designed the mask to have 1480x1480 pixels with transmission-absorption ratio of 1:1, and dimensions of 160mm x 160mm x 1.5mm with a feature size of 108 μm .’ The aperture size of the mask is 108 microns” refers to the mask we used in the primary CT experiment that we performed with the medical imaging

system at 80 kVp and that is described in the main text. The other sentence: “the FWHM of the autocorrelation function in the horizontal and vertical directions is 175 and 127 microns respectively”, refers to the second experiment when we worked at low photon energy, which is described in the Supplementary Materials. The primary experiment had the same beam amplification in both the horizontal and the vertical directions. We have revised the manuscript to ensure that this information is clearer to the readers:

Spatial resolution

The spatial resolution of our method can be evaluated by measuring the width of the autocorrelation function of the mask that modulates the input X-ray beam 58. The autocorrelation function of the mask we used in another, lower energy experiment is presented in Extended Data Fig.2(b), while the 1D horizontal and vertical projections are presented in Extended Data Fig.2(c) and Extended Data Fig.2(d), respectively. The autocorrelation function is nearly isotropic and the FWHM of the curve are 175 μm and 127 μm for the horizontal and vertical axes, respectively. We used this example because we did not have a high-resolution X-ray camera for high energy X-ray which would allow us to take images of the silver mask we used in the main experiment.

Another method for assessing spatial resolution involves the use of Fourier Ring Correlation (FRC). FRC determines resolution by cross-correlating two distinct images of the same object across different spatial frequencies, often visualized as rings in Fourier space. When noise or artifacts significantly affect the signal, it results in a low correlation, revealing constraints in measurement resolution.

The FRC plot for the main experiment is presented in Fig. 4o, and the analysis suggests that the resolution of the primary experiment is approximately 500 μm . (page 13 first paragraph).

Reviewer comment (17):

Figure legend for Extended Data Fig.4 is hard to be understood.

Our reply to comment (17):

We have revised the figure legend of Extended Data Fig.4 to enhance its clarity:

Extended Data Fig. 4 | Reconstruction images of the object with 3000 realizations. (a) Reconstruction of the object which was sampled with the detector we positioned on the left side of the object. (b) Reconstruction of the object which was sampled with the detector we positioned on the right side of the object (c) Average of (a) and (b). This comparison shows that different detector positions result in different image reconstructions due to self-absorption and scatter of the object. These results provide valuable insights on the density and composition of the object.

Summary: we thank the reviewer for carefully reviewing our manuscript and for the helpful comments that made our manuscript much clearer and our arguments more compelling. We believe that we addressed all the concerns of the reviewer and hope he/she will positively assess the revised version.

A detailed response to the Reviewer 3

The third reviewer recognizes the significance of the results presented in the manuscript and indicates a willingness to recommend publication, provided that the points raised by the reviewer are duly taken into consideration.

Our reply: we sincerely appreciate the reviewer's positive assessment of our manuscript and his/her willingness to recommend its publication. Below, we provide a detailed response to the comments of the reviewer.

Reviewer comment (1):

The basic geometry, in Fig. 2 of the present paper, may be compared to the proposal in Fig. 5b of A. M. Kingston et al., "Neutron ghost imaging", Physical Review A 101, 053844 (2020). While the latter paper considers neutrons rather than x-ray radiation as the penetrating probe, the key concept appears to be similar, namely that scattered radiation may be used to perform tomography via computational ghost imaging employing penetrating radiation. This paper by Kingston et al. should therefore be cited, and very briefly commented upon in the revised manuscript.

Our reply to comment (1):

We thank the reviewer for making this important suggestion. The approaches are indeed similar. Following the reviewer's recommendation, we have added a citation to the paper by A. M. Kingston et al. We have also added a brief comment on the paper as follows: "we apply this technique to incoherent (Compton) scattered X-ray radiation. This approach is akin to the method proposed by A. M Kingston et al. for imaging of scattered neutrons⁴²." (page 3 first paragraph)

Reviewer comment (2):

Line 57: When first mentioning computational ghost imaging, it would be useful for many readers to give a suitable reference, e.g. to J. H. Shapiro, "Computational ghost imaging", Phys. Rev. A 78, 061802(R) (2008). This would be of benefit to readers who are not already familiar with the concept of computational ghost imaging.

Our reply to comment (2):

This is another valuable suggestion that we are pleased to incorporate. Following this remark, we have included 3 additional references including the reference that the reviewer suggested in the manuscript: " We combine computational ghost imaging (CGI)²⁶⁻²⁸"(page 2 paragraph 3).

Reviewer comment (3):

Line 66, references 30-33 are cited for x-ray ghost imaging using transmitted light. Reference 30 (published in 2016) should be augmented by a second relevant paper published in the same year, namely H. Yu et al., "Fourier-transform ghost imaging with hard X rays," Phys. Rev. Lett., vol. 117, no. 11, 2016, Art. no. 113901. Adding this citation would then mean that both of the original 2016 x-ray ghost-imaging papers were cited.

Our reply to comment (3):

We appreciate the suggestions and have included the citation for the suggested paper by H. Yu et al., "Fourier-transform ghost imaging with hard X rays," Phys. Rev. Lett., vol. 117, no. 11, 2016, Art. No. 113901" in the new version of the manuscript.

Reviewer comment (4):

Figure 3 would benefit from a spatial scale bar, similar to that which is given in Fig. 4. Also, regarding Fig. 3, is there some way of more strongly indicating that the finer isosurface features in column 2 (relative to the smoother isosurfaces in column 1) are indeed genuine fine-detail information rather than artefact, beyond the indications that are currently given in the main text?

Our reply to comment (4):

We greatly appreciate the reviewer's insightful comment. As per your suggestion, we have now included a scalebar in Fig. 3. In addition, the scatter reconstruction reveals finer details on the surface of the object Due to its resilience to image degradation caused by scattering, which added noise to the transmission CT images. We have revised the manuscript to improve the clarity "Due to its resilience to image degradation caused by scattering, the scatter reconstruction reveals finer details on the surface of the object when compared to the transmission reconstruction, demonstrating the superior ability of our method to capture intricate object features." (page 6 last paragraph)

Reviewer comment (5):

How was the tomogram combination performed, e.g. in passing from columns 1 and 2 to column 3 in Fig. 4? Just a little more detail, on this point, would be helpful for the reader's understanding of the paper. Also, why are four significant figures needed for the spatial-scale-bar label, in Fig. 4?

Our reply to comment (5):

We thank the reviewer for pointing out that our reconstruction procedure of the combined tomograms was not clear enough. Following this comment, we have revised the manuscript and now it reads: “To validate our method and to benchmark its efficiency, resolution, and sensitivity against the standard direct transmission CT reconstruction, we reconstructed 3D images of a bone using signals obtained from transmission negative images, scattering reconstruction images, and their average.” (page 6 paragraph 2)

Reviewer comment (6):

Lines 151-154 state that “unlike previous works on 3D reconstructions using single-pixel detectors with visible light [41,42], which recovered only the surface gradients to derive the 3D surface of the object, we reconstructed a 3D volume that contains information about both the internal parts and the surface of the object”. In this context, the following paper should certainly be cited and briefly commented upon, since it employed computational ghost imaging to perform genuinely 3D reconstructions, albeit with transmitted rather than scattered x-ray radiation: A.M. Kingston et al., "Ghost tomography", *Optica* 5, 1516-1520 (2018).

Our reply to comment (6):

We have added the citation and commented as follows: “It is important to note that unlike previous works on 3D reconstructions using single-pixel detectors with visible light ^{48,49}, which recovered only the surface gradients to derive the 3D surface of the object, we, like A. M. Kingston et al. which achieved GI tomographic reconstruction of transmitted radiation with synchrotron radiation ⁵⁰, reconstructed a 3D volume that contains information about both the internal parts and the surface of the object with a tabletop X-ray setup.”(page 7 paragraph 2)

Reviewer comment (7):

Line 15: Perhaps “opaque instruments” should be something more like “other opaque samples”?

Our reply to comment (7):

We have revised the abstract in accordance with the reviewer’s suggestion: “X-ray imaging is a prevalent technique for non-invasively visualizing the interior of the human body and other opaque samples”

Reviewer comment (8):

Lines 68-71: The reference style changes from numbers in superscript font, to numbers in non-superscript font in round brackets. Also, are the cited references correct, in this paragraph?

Our reply to comment (8):

We appreciate the reviewer for bringing these errors to our attention. We have added the citation and commented as follows: “It is important to note that unlike previous works on 3D reconstructions using single-pixel detectors with visible light^{46,47}” (page 7 paragraph 2)

Reviewer comment (9):

Lines 173-176 state that “Since our method relies on scanning, the measurement time is proportional to the number of iterations. However, to achieve high image quality, even with clever reconstruction algorithms, the quality of the image is degraded when the number of iterations is much lower than the number of pixels”. Is this the same use of the term “iteration”, in the sense of "iterations of an image-processing code", that is given in line 89? I ask this because the words "the measurement time is proportional to the number of iterations" might be read (or misread?) as stating that the experimental measurement time is directly proportional to the number of computational iterations required to analyse the data? Please clarify the text here.

Our reply to comment (9):

We appreciate the reviewer for bringing this potential confusion to our attention. To mitigate any misunderstanding, we have replaced the term “iterations” with “samplings” in the manuscript, as follows: “Since our method relies on scanning, the measurement time is proportional to the number of samplings taken.” (page 7 last paragraph)

Reviewer comment (10):

Line 310 speaks of a “unique process”. Why is the indicated process unique?

Our reply to comment (10):

We appreciate the reviewer’s question. The process of the mask creation is considered unique because it is based on a patented technology and is done only by the Nano Dimension company.

Reviewer comment (11):

Line 316 speaks of a “low-resolution flat panel detector”. What is the resolution for this particular detector?

Our reply to comment (11):

Thank you for your valuable comment. In response, we have added an explanation to ensure that this critical information is clear to the readers: “The resolution of the images obtained from the transmission data, which we obtained by our flat panel detector is expected to be approximately 500 μm .”. (page 11 paragraph 2)

Reviewer comment (12):

Line 330, “Far-field” should not begin with a capital letter.

Our reply to comment (12):

We have amended to “far-fleld” as suggested.

Reviewer comment (13):

Line 334, “#learning rate” should not be indented differently to the other comments.

Our reply to comment (13):

We have aligned all comments.

Reviewer comment (14):

Line 341, “intel” should be “Intel”.

Our reply to comment (14):

We have corrected the typo.

Reviewer comment (15):

Line 353, a little more specific detail regarding the nature of the “computational considerations” would be useful for the reader.

Our reply to comment (15):

We are grateful for this comment and in response have included a detailed paragraph that explains the “computational considerations” (page 12 under simulation details):

Radiation source – In the laboratory set-up, the radiation field impacting the object is primarily defined by the X-ray tube and the mask. To achieve high resolution CGI we employed masks with thousands of fine features. However, representing these masks as physical objects in MC programs and changing them for each sampling is computationally demanding and time consuming. This is because MC simulations track each primary photon separately and record the interactions along its path. To circumvent complexities associated with modeling thousands of masks and their interactions, including self-absorption, self-scatter, recoil, and energy loss, we adopted a more efficient approach for modeling the radiation source. In our simulation, we defined a radiation source that represents the field immediately after its interaction with the masks, positioned 5 cm before reaching the object. The 80 kVp energy spectrum was obtained with the SPEKTR 3.0 program⁵⁶ using the TASMICS algorithm. This spectrum was generated for a Fewell tube with a tungsten anode and inherent 1.6 mm Al filtration, without any additional filtration.

Reviewer comment (16):

Line 398, should “is shown” be “, as shown”?

Our reply to comment (16):

We have revised the manuscript in accordance with the reviewer’s comment.

Reviewer comment (17):

Line 402, is the word “in” missing after “function”?

Our reply to comment (17):

We have added the word “in” after the word “function” in line 402.

Reviewer comment (18):

Line 422, “could” should probably be “would”.

Our reply to comment (18):

We have revised the term “could” to the term “would” in the new version.

Reviewer comment (19):

Extended data, Fig. 2, “determanation” is incorrectly spelled. Extended data, Fig. 3, “experimentl” is incorrectly spelled. Similarly for Extended data, Fig. 4, whose caption has four spelling errors.

Our reply to comment (19):

We have amended the manuscript to address the comments and remove typos.

Reviewer comment (20):

The references need to be properly proofread. For example, weblinks appear in place of the journal name in references 2 and 4, the journal name in reference 5 is not abbreviated, reference 31 has an asterisk that should not be there, reference 44 is missing its page number, reference 47 is missing journal details and page number, reference 50 is missing its article number, etc.

Our reply to comment (20):

We have amended the manuscript to address the comments and remove typos.

Summary: We express our sincere gratitude to the referee for taking the time to meticulously review our manuscript and for providing valuable insights that have significantly enhanced the clarity of our manuscript. We have carefully addressed all concerns raised by the reviewer and have meticulously revised the manuscript accordingly.

Reviewers' comments:

Reviewer #1 (Remarks to the Author):

The authors have responded to all my comments. I appreciate that and do not have further comments.

Reviewer #2 (Remarks to the Author):

All the questions were well answered, and the revised manuscript became more understandable.

However, there are still points to be addressed.

1. In the manuscript the projection angles were selected as 0, 180, 90, 270, etc. Is there a standard for the choice of the projection angles. If the selection of 28 projection angles in the range of 0-90 degrees instead of 0-360 degrees has any effect on the reconstruction quality.

2. No citations related to the FRC graph and the half-bit graph are included to help the readers' understanding.

3. Briefly describe the process of finding an optimal configuration φ^* for the neural network. And what kind of training data is needed to help find the optimal parameters φ^* .

4. Is the scattered signal obtained only from the detector at 90° in the experiment? In the section of Methods, simulation details show the scattered signals are from the detectors at different angles. Is the scattered signal used in the CGI reconstruction from the detector at a single angle or the sum of detectors at all the angles?

5. In the manuscript "Scatter + Transmission" denotes the average of the transmission negative images and scattering reconstruction images. Due to the weak scattering signals resulting in the scattering reconstruction images with small numerical values, is the "Scatter + Transmission" image obtained through the weighted average between the transmission negative images and scattering reconstruction images? If so, is there a requirement for weights?

6. In the last sentence "However, to achieve high image quality, even with clever reconstruction algorithms, the quality of the image is degraded when the number of iterations is much lower than the number of pixels" of Page 7, the letter "iterations" is advised to be changed to samplings to avoid misreads.

7. At the bottom of Page 7, "the quality of the image is degraded when the number of iterations is much lower than the number of pixels." here "iterations" should also be replaced by "measurements".

8. Typos: at Extended Data Fig.2, "Spatial resolution assesment"; at Extended Data Fig.3, "7 trasmission image".

Reviewer #3 (Remarks to the Author):

All of my comments have been carefully taken into account in preparing the revised version of the manuscript. I therefore strongly recommend that this excellent paper be accepted without further review.

High-resolution computed tomography with scattered X-ray radiation and a single pixel detector

A. Ben Yehuda¹, O. Sefi¹, Y. Klein¹, R. H. Shukrun^{1,3}, H. Schwartz¹, E. Cohen², and S. Shwartz^{1*}

Detailed response to the comments of the Reviewers

A detailed response to the Reviewer 2

We thank Reviewer 2 for the valuable assessment of our manuscript. In response to the reviewer's insightful comments, we have addressed all of them as outlined below:

Reviewer comment (1):

In the manuscript the projection angles were selected as 0, 180, 90, 270, etc. Is there a standard for the choice of the projection angles. If the selection of 28 projection angles in the range of 0-90 degrees instead of 0-360 degrees has any effect on the reconstruction quality.

Our reply to comment (1):

Thank you for your question regarding the selection of projection angles in our manuscript. The choice of projection angles is a crucial aspect of any computed tomography (CT) reconstruction, and we appreciate the opportunity to provide further clarification.

In our study, we used a full 360-degree rotation, which is associated with obtaining a more complete and accurate representation of the imaged object compared to limited angle acquisitions. It helps reducing artifacts and enhances the ability to visualize and analyze fine structures.

However, we acknowledge the potential impact of this specific choice on reconstruction quality. Using 28 projection angles within the range of 0-90 degrees could have the potential to introduce artifacts and degradation of the image quality. This limitation might impact the accuracy and fidelity of the reconstructed images compared to a more comprehensive angle range.

Reviewer comment (2):

No citations related to the FRC graph and the half-bit graph are included to help the readers' understanding.

Our reply to comment (2):

Thank you for bringing attention to the absence of citations related to the FRC graph and the half-bit graph in our manuscript. We have addressed this concern by incorporating relevant citations that provide context and support for the interpretation of these graphical representations. We

believe these additions will significantly enhance the readers' understanding of the respective graphs.

We appreciate your diligence in reviewing our work and therefore added the citations to the manuscript:

“To further explore the ability of our method to resolve fine details, we evaluated the reconstruction resolution by measuring the cutoff frequency of the Fourier ring correlation (FRC)⁵⁰⁻⁵² of two different, 3000 samplings reconstructions (Extended Data Fig.7).” (page 7 paragraph 2)

Reviewer comment (3):

Briefly describe the process of finding an optimal configuration φ^* for the neural network. And what kind of training data is needed to help find the optimal parameters φ^* .

Our reply to comment (3):

Thank you for your insightful comment regarding the process of finding an optimal configuration φ^* for the neural network and the type of training data required to determine the optimal parameters φ^* .

The optimization of φ^* was performed utilizing established methods implemented in the open-source code. This includes considerations such as hyperparameter tuning, architecture selection, and other relevant optimization strategies. The specific details of this process, including the training data used, can be found in the referenced code and associated documentation.

Reviewer comment (4):

Is the scattered signal obtained only from the detector at 90° in the experiment? In the section of Methods, simulation details show the scattered signals are from the detectors at different angles. Is the scattered signal used in the CGI reconstruction from the detector at a single angle or the sum of detectors at all the angles?

Our reply to comment (4):

We appreciate your inquiry and would like to clarify that when we measured the scattered radiation for the CT experiment the detector was mounted at 90°. This information is indicated on page 5. The simulation details in the Methods section, mentioning scattered signals from detectors at various angles, are aimed at providing a comprehensive overview of the simulation setup.

We hope this clarification addresses your concern.

Reviewer comment (5):

In the manuscript “Scatter + Transmission” denotes the average of the transmission negative images and scattering reconstruction images. Due to the weak scattering signals resulting in the

scattering reconstruction images with small numerical values, is the “Scatter + Transmission” image obtained through the weighted average between the transmission negative images and scattering reconstruction images? If so, is there a requirement for weights?

Our reply to comment (5):

Thank you for your comment. We appreciate the opportunity to clarify that the images, including both the transmission negative images and the scattering reconstruction images, are normalized in our study. The normalization is applied to ensure a consistent and comparable scale across the images, considering the potential variations in intensity.

While normalization was implemented in our methodology, we recognize the importance of explicitly mentioning this detail in the manuscript to enhance clarity. We therefore clarified the manuscript accordingly:

“To validate our method and to benchmark its efficiency, resolution, and sensitivity against the standard direct transmission CT reconstruction, we reconstructed 3D images of a bone using signals obtained from transmission negative images, scattering reconstruction images, and their normalized average. The results of these reconstructions are presented in Figs. 3-4.” (Page 6 paragraph 2)

Reviewer comment (6):

In the last sentence “However, to achieve high image quality, even with clever reconstruction algorithms, the quality of the image is degraded when the number of iterations is much lower than the number of pixels” of Page 7, the letter “iterations” is advised to be changed to samplings to avoid misreads.

Reviewer comment (7):

At the bottom of Page 7, “the quality of the image is degraded when the number of iterations is much lower than the number of pixels.” here “iterations” should also be replaced by “measurements”.

Our reply to comment (6 and 7):

Thank you for your keen observation and suggestion regarding the potential misinterpretation in the last sentence on page 7. We have duly addressed the concern and modified the term "iterations" to "samplings" in the revised version of the manuscript.

Reviewer comment (8):

Typos: at Extended Data Fig.2, “Spatial resolution assesment”; at Extended Data Fig.3, “7 trasmission image”.

Our reply to comment (8):

Thank you for pointing out the typos in Extended Data Fig. 2 ("Spatial resolution assessment") and Extended Data Fig. 3 ("7 transmission image"). We have made the necessary corrections to address these issues in the revised version of the manuscript.

Summary: We sincerely appreciate the reviewers' thoughtful and detailed review of our manuscript. His/her insightful comments and suggestions have been invaluable in enhancing the clarity and quality of our work.

REVIEWERS' COMMENTS:

Reviewer #2 (Remarks to the Author):

The comments and questions are addressed accordingly in details. I can recommend its publication in Communications Engineering now.